# The efficacy of interventions in the workplace promoting exercise and a healthy diet among shift workers: A systematic review

Francielle Lopes dos Reis[1,*], Julio Cesar Ferreira Bertoloto[2],
Ticiana da Costa Rodrigues[3], Sheila de Castro Cardoso Toniasso[4],
Camila Pereira Baldin[4‡], Juliana Barros Rodrigues[2‡], Dvora Joveleviths[5‡],
Maria Carlota Borba Brum[5]

1 Outpatient Nursing Service, Hospital de Clínicas de Porto Alegre, Porto Alegre, Rio Grande do Sul, Brazil, 2 Federal University of Rio Grande do Sul, School of Medicine, Porto Alegre, Rio Grande do Sul, Brazil, 3 Federal University of Rio Grande do Sul, Department of Internal Medicine, Porto Alegre, Rio Grande do Sul, Brazil, 4 Occupational Medicine Service, Hospital de Clínicas de Porto Alegre, Porto Alegre, Rio Grande do Sul, Brazil, 5 Federal University of Rio Grande do Sul, Department of Social Medicine, Porto Alegre, Rio Grande do Sul, Brazil

☯ These authors contributed equally to this work.
‡ These authors also contributed equally to this work.
* flreis@hcpa.edu.br

## Abstract

### Introduction

Chronic non-communicable diseases (CNCDs) are a major public health concern, with significant impacts on quality of life and health costs. Shift work is a risk factor associated with these diseases, since it interferes with circadian rhythms and physiological processes, and can lead to circadian desynchronization and sleep deprivation. Given this scenario, the workplace is recognized by the WHO as a strategic environment for promoting health and preventing CNCDs.

### Objective

To analyze the scientific evidence regarding the efficacy of interventions to promote healthy eating and/or physical activities among shift workers.

### Methods

Systematic review protocol scientific databases in the field of health: MEDLINE (via PUBMED), Excerpta Medica Database (Embase), Latin American and Caribbean Literature in Health Sciences (LILACS), Web of Science and Scientific Electronic Library Online (SciELO), between January 2013 and December 2023, and was registered in the database for the study of the systematic review PROSPERO, under number: CRD42024517563. The risk of bias was analyzed according to the assessment tool, RoB 2.0 (Revised Cochrane risk-of-bias tool for randomized trials), two of

**Data availability statement:** All relevant data are within the paper and its Supporting Information files.

**Funding:** Research and Events Incentive Fund - FIPE from Hospital de Clínicas de Porto Alegre (2022-0618).

**Competing interests:** The authors have declared that no competing interests exist.

the studies were evaluated using the risk of bias tool by the Non-randomized Studies – of Interventions (ROBIN I).

## Results

The electronic search resulted in 2361 relevant articles based on the database search. After removing duplicates and articles that did not meet the inclusion criteria, 366 articles were identified. Thirteen articles were selected for full-text review, and 7 articles were included.

## Discussion

The selected studies show that health interventions in the workplace, although with differences in the types of interventions and populations, have favorable results. Strategies aimed at nutritional support and physical activity, with the use of technologies such as motivational messages, have shown a positive impact, which is amplified when it is possible to involve workers and adapt them in the workplace. The diversity in study designs offers a broad perspective, but the variability in research methods also brings significant challenges for comparability, which justifies the decision not to carry out a meta-analysis. In addition, most studies focus on short-term interventions and outcomes, which may not adequately reflect the long-term health benefits or risks associated with shift work.

## Conclusion

The diversity of interventions suggests that there is no single solution to promote health at the workplace. The strategies can be adapted to the specific needs and contexts of the workers and working environments. The adherence of the managers is a way of reinforcing the importance of preventative actions and allows a better adaptation of the organizational context to these activities. The lack of long-term follow-up and continued adherence are challenges that need more investment and organizational policies to ensure the effectiveness of the actions.

## Introduction

The World Health Organization considers (WHO) noncommunicable diseases (NCDs) public health problems, characterized by the set of pathologies with different causes and related risk factors, may result in incapacities, reduced quality of life and increased costs of healthcare and social security, preventative measures to fight them have been constantly investigated. The repercussions of these diseases go beyond the individual sphere, and may also affect the socioeconomic scene, and there is evidence of association between chronic diseases and occupational risk factors, especially shift work [1].

Shift work to expose the workers to alterations of their physiological functions that influence and have repercussions on their health and become a potential risk

for chronic diseases [2–4]. In humans, circadian rhythms are synchronized mainly by photic stimuli (light), although non-photic rhythms also influence the biological clock, such as eating times, physical activity and social interactions [5,6]. Thus, among shift workers, biological rhythms are not synchronized with external cycles. The pathological mechanisms by which shift work contributes to damage to health are not well established, but two of them have been corroborated by the evidence: circadian desynchronization and sleep deprivation. The biological rhythms affected by desynchronization in night workers are mainly sleep, alertness, performance in tasks that require attention, as well as changes in metabolic and hormonal rhythms [7].

Blood glucose homeostasis is associated with central circadian rhythmicity as well as peripheral oscillators. During periods of activity, blood sugar levels are determined mainly by nutrient intake. During rest and fasting, endogenous hepatic glucose production occurs and glucose levels are maintained within a relatively narrow range [8]. This also includes intestinal hormones, associated with the regulation of hunger and satiety, as well as gastrointestinal functioning and the kinetic properties of the stomach. Ghrelin stimulates food intake, while leptin promotes satiety. The production of these hormones can be interfered with by sleep deprivation and food intake, leading to an internal desynchronization of the circadian rhythmicity of satiety-related peptides, especially leptin, causing an imbalance between energy intake and expenditure [9–12].

Given this scenario, workplaces were recognized by the WHO as environments with a potential to promote healthy lifestyles and environments favorable to well-being. They are privileged places for the prevention and promotion of health since they cover a large part of the active population and can implement actions to protect the worker [13,14]. Time spent at work is a key opportunity or health promotion, affecting a segment of the adult population that spends a large proportion of waking hours at work.

There is evidence of the benefits of supporting health promotion activities in the workplace with the aim of maintaining the capacity, health and quality of life of workers, with potential impacts on health care costs and on the productivity, competitiveness and sustainability of companies, communities and even national economies, considering both the risks in work environments and other factors such as ageing and the need to adapt the workplace of people with chronic conditions [15].

Numerous opportunities exist for health promotion and chronic disease prevention initiatives in the workplace, including (a) health screening, (b) tobacco cessation activities, (c) the promotion of healthy food choices and weight loss, (d) physical activity, and (e) routine vaccinations [16]. However, the efficacy of these interventions has not yet been fully established, and it is necessary to perform a broad review of the evidence available to evaluate whether the interventions present favorable results regarding the living habits adopted by the workers, especially when it is a matter of diet and exercises essential to prevent NCDs.

Workplace interventions have objectives related to the promotion and reduction of health care costs. In this sense, studies have been carried out to identify the effectiveness of these interventions, showing positively significant effects in relation to physical activity and nutrition, related to self-reported behavior, but without significant results in relation to physical health measures, rates of medical diagnoses or use of health services [17].

Thus, the purpose of this systematic review is to analyze the existing studies on interventions at the workplace turned towards the promotion of physical exercises and a healthy diet among shift workers. By summarizing and evaluating this evidence, we hope to supply important perceptions to inform the development of effective policies and practices to improve the health and well-being of the professionals.

## Materials and methods

The systematic review following the guidelines designed by PRISMA (Preferred Reporting Items for Systematic Reviews and Meta-Analyses) [18] to respond to a research question based on the PICO (Population (shift workers), Intervention (interventions for promoting physical exercise and healthy diet), Control (not aplicable) and Outcomes (efficacy workplace interventions) strategy: "What is the efficacy of the interventions to promote a healthy diet and physical activity at the

workplace in shift workers?" This review was registered in the database for the study of the systematic review PROSPERO, under number: CRD42024517563, on March 7, 2024.

## Eligibility criteria

The following criteria were applied in the bibliographic search for published research articles: (1) published between 2013 and December 2023; published in English, Portuguese or Spanish language; (3) concerning interventions in the work environment; (4) describing interventions to promote health related to physical activity and/or diet; (5) describing the result of the intervention; (6) citing work shifts. No limitations were established regarding the gender of the subject, workplace size, number of employees, control group, health risks evaluation (HRA), type of intervention, environmental, educational and multicomponent. We included intervention studies with different designs, including randomized clinical trials in order to pick up the greatest possible number of studies relevant to study the impact of the interventions on the work environment.

## Exclusion criteria

Were excluded observational articles, duplicates, theses, integrative reviews, narrative reviews, systematic reviews with or without meta-analysis, preclinical studies, ecological studies, cost-effectiveness analyses, books, journals, commentaries, letters to the editor, book chapters; political documents, statements of consensus, study protocols or articles that did not concern the object of research.

## Search strategy

The searches were performed from January to December 2023, in databases: MEDLINE (via PubMed), Excerpta Medica Database (Embase), Literatura Latino-Americana e do Caribe em Ciências da Saúde (LILACS), Web of Science and Scientific Electronic Library Online (SciELO). Two independent reviewers performed the searches and a third reviewer was requested for the cases in which there were divergences. The search strategy is shown below, in each database and their respective selection filters. The boolean operators AND and OR were utilized as a strategy for crossing information among the descriptors:

**PUBMED:** (occupational health) AND (intervention) NOT "Animals"[Mesh] Agrochemicals); Shift Work Schedule"[MeSH Terms] AND "Healthy Lifestyle"[MeSH Terms]; dietary change"[MeSH Terms] AND "workplace"[MeSH Terms] AND "health promotion", habits"[MeSH Terms] AND "occupational health"[MeSH Terms] AND "health promotion", (("Exercise"[Mesh]) AND "Shift Work Schedule"[Mesh]) AND "Workplace"[Mesh], (("Occupational Health"[Mesh]) AND ("Clinical Trials as Topic"[Mesh] OR "Clinical Trial" [Publication Type])) AND "Health Promotion"[Mesh], ("Health Promotion"[Mesh]) AND "Shift Work Schedule"[Mesh], ("Health Promotion"[Mesh]) AND "Workplace"[Mesh]

**Embase:** ('health behavior'/exp OR 'health behavior') AND ('occupational health'/exp OR 'occupational health') AND ('workplace'/exp OR workplace) AND ('shift work'/exp OR 'shift work'); habits AND occupational AND health AND promotion result; shift work AND work AND 'health promotion' AND 'workplace'; physical activity' AND workplace AND 'occupational health' AND 'shift work'.

**LILACS**: Occupational AND health and shift work; physical activity AND workplace AND occupational health; Exercise AND Workplace

**Web of Science**: Dietary change and workplace; shift work AND work AND 'health promotion'; occupational AND health AND promotion AND shift work; physical activity and occupational health and workplace

**SCIELO**: Shift work AND work AND 'health promotion'; dietary change and workplace; occupational AND health AND promotion AND shift work; physical activity and occupational health and workplace; (physical activity) AND (occupational health); (shift work) AND (intervention study).

## Selection of the studies and data extraction

The Rayyan app [19] (intelligent systematic review) was utilized to manage the studies. Two authors independently selected the titles and abstracts of all the studies according to the eligibility criteria. The reviewers were not blinded to the authors, institutions or periodicals in which the studies were published. The abstracts that did not supply sufficient information for eligibility were later inspected, independently, using the full text evaluated by the same authors. The divergences were solved by an independent third author. The following information was extracted from the full text and included: title of article, author/year, country, design, population characteristics – mean age/sex, recruitment strategies, objective of the study, inclusion and exclusion criteria, type of intervention, duration, evaluation of the intervention and results.

## Outcomes of interest

Based on the publications that show changes in diet and level of physical activity taken up by the worker in different work shifts, outcomes considered of interest in this review wee aspects involving general well-being, quality of life, circadian sleep rhythm, cognitive performance, measures of blood pressure, laboratory tests, body composition, anthropometric data, muscle strength, influence on performance at work, adverse events and dropping out.

## Evaluation of bias

The Cochrane Collaboration Tool RoB 2.0 (Revised Cochrane risk-of-bias tool for randomized trials) was used to analyze the bias of randomized studies [20] in the evaluation, two of the studies were evaluated using the risk of bias tool by the Non-randomized Studies – of Interventions (ROBIN I) [21]. Two of the authors carried out the evaluation using the tool, and the disagreements were re-evaluated by two other authors.

## Results

Surveys of the databases identified 2,361 studies, 110 of which were duplicates and were excluded. After the duplicate articles were excluded, 1885 articles were excluded because they did not correspond to the predetermined inclusion criteria, and 366 articles were pre-selected for reading the title and abstract, 353 were excluded, thus rendering a total of 13 articles eligible for full reading. Later, 6 articles were excluded because they did not fit the eligibility criteria. Thus, 7 articles were included in this review. Fig 1 shows the flowchart that summarizes the selection process in this systematic review.

The main characteristics of the studies are shown in Table 1. In this scenario of evidence, the population selected was composed of health professionals, long distance truck drivers, policemen and mining and industry workers, with a predominance of women among health professionals and men among other activities, aged between 30 and 48.4 years. Three of the studies selected are randomized prospective studies, one retrospective study, one cluster-controlled trial, one non-blind randomized/controlled trial, and one pre and post test pilot intervention study. The studies were carried out in Australia (02), EUA (01), Japan (01), Ireland (01), Taiwan (01) e United Kingdom (01). The recruitment strategies used involved sending e-mail, electronic newsletters and posters at the workplace; other strategies included convenience recruitment through information card and prior meeting, pre-evaluated by telephone to determine eligibility and personal invitations.

The duration of the interventions in the studies was variable, ranging from 7 to 24 weeks, and only 4 studies [23,25,27,28] presented follow-up measures after the interventions, with results favoring the continuity of behavioral changes up to 96 weeks after the interventions. The use of combined strategies (physical activity and nutrition), health coach support, definition individual goals and self-monitoring were present in these studies. The information regarding the types of intervention adopted, time of application and results of interest for the selected studies are detailed in Table 2.

In all studies, favorable results were found for the interventions performed. The use of strategies to promote the practice of physical exercise and healthy eating were discussed in 02 studies [24–27], while the other studies addressed only

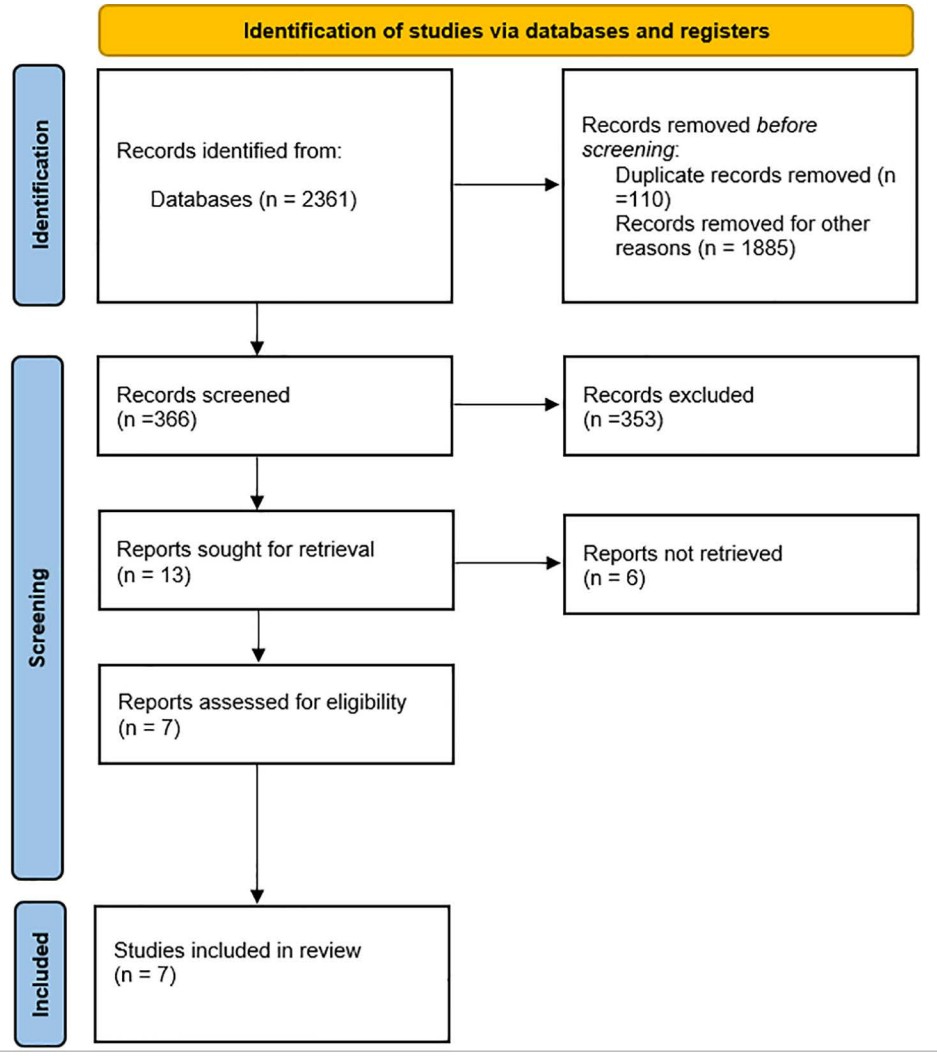

**Fig 1. Prisma flowchart for article identification.**

one of the strategies (03 studies on physical activity and 02 on nutrition). Geaney et al focused exclusively on issues related to diet, while Kuehl et al associated the analysis of other aspects of health promotion (sleep, stress, well-being, tobacco, alcohol and ideal body weight) using the method Structured Health Intervention for Truckers (SHIFT).

The study conducted by Monnaatsie *et al* utilized the Reach, Effectiveness, Adoption, Implementation, and Maintenance (RE-AIM) structure to evaluate the intervention program that involves sensibilization and information to the participants before recruitment, techniques on the changes of behavior, feedback on physical activity, planning of actions and definition of specific goals [29], besides sending personalized text messages during the activities, according to the participant's shift. Together with this model, the facilitating factors or barriers to the implementation and maintenance of the program to promote physical health were evaluated. Torquati L *et al*, also utilized this model directed toward physical activity and diet among nurses. Although it involved a different population, it was possible to identify the authors satisfaction with the use of this method regarding the organization and evaluation of the intervention.

**Table 1. Main characteristics of the studies selected.**

| Author/Year | Title of the Article | Country | Study Design Population | Objective | Inclusion/Exclusion Criteria |
|---|---|---|---|---|---|
| Monnaatsie et al., 2023 [22] | The Feasibility of a Text-Messaging Intervention Promoting Physical Activity in Shift Workers: A Process Evaluation | Australia | RCT N = 60 workers in mining company > 39,3 y males 53% | Evaluation of the process of intervention to promote health through text messages. | Inclusion: Physical activity practice; download application, English fluency. Exclusion: Not specified. |
| Clemes et al., 2022 [23] | The effectiveness of the s SHIFT: a RCT | United Kingdom | RCT N = 382 long distance truck drivers (183 SHIFT arm and 199 controls > 48,4 y males 99% | Evaluation of the effectiveness of the SHIFT, compared to the usual care in increasing the physical activity and the reduction of sedentary time among the heavy long-distance vehicle drivers. | Inclusion: Voluntary participation. Exclusion: Clinically diagnosed heart disease, hemophilia, blood-borne viruses or significant limitations of mobility. |
| Cheng et al., 2022 [24] | Effectiveness and response differences of a multidisciplinary workplace health promotion program for healthcare workers | Taiwan | Retrospective study N = 36 HCWs Median 36 y women 93% | Evaluation of the impact and the differences in the response to an intervention turned to nutrition and physical exercises. | Inclusion: 18–64 years, voluntary participation. Exclusion: History of stroke, coronary artery disease or other major vascular diseases. |
| Geaney et al., 2016 [25] | The effect of complex workplace dietary interventions on employees' dietary intakes, nutrition knowledge and health status: a cluster-controlled trial | Ireland | Cluster controlled trial N = 517 workers in 4 industries 30 - 44 y male predominance | Evaluation of the comparative efficacy of environmental dietary change at the workplace and an intervention of nutritional education alone or combined versus a workplace with control (FCW). | Inclusion: Full-time permanent workers, eating their main meal at work daily. Exclusions: Not working full-time, regularly traveling for work, long-term sick leave, pregnancy or diet program. |
| Matsugaki et al., 2017 [26] | Effectiveness of workplace exercise supervised by a physical therapist among nurses conducting shift work: A RCT | Japan | Non-blind randomized and controlled N = 30 nurses > 24.7 y female | Evaluation of the effectiveness of supervised exercise among nurses who do shift work to promote health. | Inclusion: Age 20–40 years, shift work full time. Exclusion: medical problems in outpatient treatment. |
| Torquati et al., 2017 [27] | Changing Diet and Physical Activity in Nurses: A Pilot Study and Process Evaluation Highlighting Challenges in Workplace Health Promotion | Australia | Pre and post test pilot intervention study N = 47 nurses > 41,4 y female 87% | Evaluate the main factors for the implementation and context of an intervention regarding diet and physical activity at the workplace | Inclusion criterion: Full time or partial night work. Exclusion: Non-controlled hypertension, diabetes or unstable angina, orthopedic and neurological limitations, pregnancy or surgery planned. |
| Kuehl et al., 2017 [28] | The SHIELD Study: Mixed Methods Longitudinal Findings | United States | RCT N = 408 police officers and support staff > 41,7 y male predominance | Seven goals: healthy daily exercise and nutrition, achieving the ideal body weight, reducing stress and improving sleep, reducing the excessive use of alcohol and stopping smoking. | No inclusion and exclusion criteria. |

Reis, F.L *et al*, 2024. Caption: BMI: body mass index; FCW: Food Choice at Work; HCWs: healthcare workers; RCT: cluster randomized controlled trial; SHIELD: Safety & Health Improvement: Enhancing Law Enforcement Departments; SHIFT: Structured Health Intervention For Truckers. * Median.

Geaney used an intervention design using another model called the Medical Research Council (MRC) that is used to develop evaluations of complex interventions. The intervention structure included 4 approaches per workplace, related to only collecting data without intervention, nutritional education (NE) and environmental modification of the diet (EMD), alone or combined [30], based on previous information among the workers, showing favorable results in the medium time.

**Table 2. Description of the types of intervention, duration and outcomes of interest of the selected evidence.**

| Author/Year | Type of Intervention | Duration of intervention | Evaluation of intervention | Results |
|---|---|---|---|---|
| Monnaatsie et al., 2023 [22] | Individual feedback, actions plans (adherence strategies and adaptation of needs), monitor physical activity, motivational text messages. Control group: Generic information. | 07 weeks | Questionnaire and use of activPAL accelerometer (heart rate, steps; level of stress and sleep). | Coverage 66%; greater organizational involvement might increase participation. The action planning approach facilitated the adoption of the measures. Participants' optimism: positive feedback continuity the program. Organizational maintenance not evaluated |
| Clemes et al., 2022 [23] | Interactive educational session group, (physical activity health coach support, self-monitoring device, actions plans, individual goals (diet and sedentary time). | 24 weeks | Occupational and health questionnaire, blood pressure and heart rate, anthropometry, capillary blood sample (lipid and glycemic profile). The duration and efficiency of sleep is evaluated by the GENEActiv triaxial accelerometer. | Potentially significant clinical difference as to the mean number of daily steps between the arms of the study at 24 weeks. There was no significant difference between the groups in 16–18 months. |
| Cheng et al., 2022 [24] | Health promotion program of exercise, nutritional consultation, behavioral education and information about the benefits of healthy habits. | 20 weeks | Basic information, anthropometry, body composition and physical health. | Significant improvement of health parameters and anthropometric measures, body composition and physical aptitude. The impact on the mean reduction of the body mass index was significantly higher in shift workers compared to those who do not work shifts. |
| Geaney et al., 2016 [25] | Intervention 4 approaches per workplace: 1) only collecting data without intervention; 2) NE; 3) EMD; 4) combined intervention NE and EMD. | Intervention and follow up at 12–16/28–36 weeks | Sociodemographic questionnaires on lifestyle, general nutritional knowledge pre and post intervention and 24-hour food diary (work days and free days), anthropometry. | Combined dietary intervention (nutritional and environmental) reduced the food intake of salt and saturated fat, improved nutritional knowledge and diminished the BMI at 28–36 weeks of follow up. |
| Matsugaki et al., 2017 [26] | Resistance and aerobic training 2 sessions/week with and without follow up with physiotherapy (intervention x control); exercise manual and motivational emails. | 12 weeks | Evaluation of aerobic aptitude, muscle strength, anthropometry, biochemical parameters and mental health (depression, psychological state and regarding habits and consciousness of exercises). | Significant improvement of muscle strength. HDL cholesterol and high molecular weight adiponectin and depressive symptoms in the control group. |
| Torquati et al., 2017 [27] | Orientations and support in nutrition and physical health, actions plans, individual goals, use of accelerometer and pedometer; access to social media groups and apps. | 12 weeks | Sociodemographic questionnaires, self-evaluation of health, self-efficacy and social support and FFQ; evaluation of anthropometric measures pre- and post-intervention and at 6 months. | Moderate success in self-monitoring and establishment of dietary goals, change in diet and in physical activity maintained at 24 weeks; low adherence to social media and insufficient social support. |
| Kuehl et al., 2017 [28] | Intervention group: 12 meetings (30 minutes) during 6 months, actions plans, individual goals. The control group: initial tests and discussion of the results. | 24 weeks | Sociodemographic questionnaires, diet, physical activity and mental and physical health; anthropometric measures, blood pressure and laboratory (lipid and glycemic profile). | Increased consumption of fruits and vegetables and healthy diet, improved quality of sleep, reduction of personal stress and reduction of the use of tobacco and alcohol, six months after intervention. At 96 weeks after intervention there was a perceptible continuity of these changes in the intervention group. |

Reis, F.L *et al*, 2024. Caption: BMI: body mass index; EMD: environmental modification of the diet; FFQ: Food frequency: High-density lipoprotein; NE: nutritional education.

The NE approaches included monthly presentations, daily nutritional information and 03 individual nutritional reviews. The EMD included: a) menu change; restriction of saturated fat, sugar and salt, b) increase in fiber, fruits and vegetables, c) discounts on the prices of whole fresh fruits, d) strategic positioning of healthier alternative foods and e) control of portion sizes.

The study conducted by Cheng et al addressed intervention among HCWs (Health care workers) involving 20-week intervention included multiple easy-to-access 90-min exercise classes, one 15-min nutrition consultation, and behavioral education.

Other studies on physical activity [23,24,25,26] used individual or group guidance associated with professional guidance and individual monitoring. Only 2 studies did not use a control group [24,25,26,27], both of which were carried out with HCWs and had a small sample size (<50 participants).

According to the assessment tool, RoB 2.0 (Revised Cochrane risk-of-bias tool for randomized trials) (Sterne et al, 2019), of the randomized trials selected in this study, 2 presented some concerns, 1 high risk and 3 low risks, as shown in Fig 2. In the evaluation, two of the studies were evaluated using the risk of bias tool by the Non-randomized Studies – of Interventions (ROBIN I), as shown in Fig 3.

## Discussion

Although the differences between the study design and population, the types of intervention, their duration and follow-up do not allow for a comparison of the results, we observed aspects that demonstrate the positive impact of these actions in the workplace. These aspects relate to the planning of strategies and individual objectives associated with nutritional and physical activity support, addressed individually or in groups, demonstrating that offering information on the topics and resources for changes in participants' lifestyles has positive results, which extend beyond the intervention period [23,25,27,28]. The use of devices to measure physical activity and access to motivational messages were resources used to optimize results in some studies and also contributed to the favorable results, although the ways of assessing the results are different, from evaluating the laboratory profile to the clinical profile, through anthropometric measurements, and changes in the participants' eating and physical routines.

The strategy structured in physical activity support [23,24] evidenced a positive impact on the health of the workers. Despite the limitations, the results were favorable and obtained by offering the support of physical trainers, equipment,

| Risk of bias domains | | | | | | |
|---|---|---|---|---|---|---|
| Estudo | D 1 | D 2 | D 3 | D 4 | D 5 | General |
| Monnaatsie, M et al 2023 | Low | Low | Low | Low | Low | Low |
| Clemes, S.A. *et al* 2022 | Low | Some Concerns | Low | Some Concerns | Low | Low |
| Cheng, K.-H. *et al*, 2022 | Not evaluated | Not evaluated | Not evaluated | Not evaluated | Not evaluated | Not evaluated |
| Geaney *et al*, 2016 | Some Concerns | Low | Low | Some Concerns | Low | Low |
| Matsugaki et al, 2017 | High | Low | Low | Low | Low | High |
| Torquati L et al, 2017 | Not evaluated | Not evaluated | Not evaluated | Not evaluated | Not evaluated | Not evaluated |
| Kuehl et al, 2017 | Low | Low | Low | Low | Low | Low |

Reis, F.L *et al*, 2024.

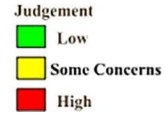

Judgement
- Low
- Some Concerns
- High

**Fig 2. Assessment of risk of bias by the Cochrane Collaboration Tool for randomized clinical trials, RoB2.** Domains: D1: Bias arising from the randomization process, D2: Bias due to deviations from intended intervention, D3: Bias due to missing outcome data, D4: Bias in measurement of the outcome, D5: Bias in selection of the reported result.

| Risk of bias domains | | | | | | | |
|---|---|---|---|---|---|---|---|
| Estudo | D 1 | D 2 | D 3 | D 4 | D 5 | D 6 | D 7 |
| Cheng, KH et al. , 2022 | Low | Serious | Moderate | Low | Moderate | Low | Low |
| Torquati L, et al., 2017 | Low | Moderate | Low | Moderate | Low | Low | Low |

Reis, F.L *et al*, 2024.

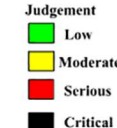

Judgement
■ Low
■ Moderate
■ Serious
■ Critical

**Fig 3. Assessment of risk of bias by the In Non-randomized Studies – of Interventions, ROBIN I.** Domains: D1: Bias due to confounding, D2: Bias in classification of interventions, D3: Bias in selection of participants into the study (or into the analysis), D4: Bias due to deviations from intended interventions, D5: Bias due to missing data, D6: Bias in measurement of the outcome, D7: Bias in selection of the reported result.

space for the physical activities and educational session. The intervention showed potential to be adopted for truckers and nurses in order to promote physical activity in these high-risk occupational groups and, consequently, improve the dietary habits and health indicators. In the case of the SHIFT program, the presence of professionals such as physical trainers is a determining factor for the intervention. A multi-professional team would be ideal to allow for a broader and more personalized approach, considering that health is not only made up of physical factors, but also psychological, social and behavioral aspects that influence lifestyle.

In Monatisse et al, the use of text messages has a positive impact on the promotion of physical activity in shift workers, acting as distance guidance and as an incentive to maintain behavioral change; it is a means of intervention with a practical and accessible approach. The phase involving the presentation and sensitization of workers was important for adherence to the proposed measures. This reinforces the need to know the profile of the workers and the characteristics of their workplace when planning health promotion activities [22].

Cheng *et al* showed that there were different effects on shift workers that presented a significantly greater mean reduction in the body mass índex, indicating a differential effect of the intervention based on the shift worker status. However, regarding sex, age or weight, no significant differences were observed in the response, suggesting that the intervention was effective independent of these variables. The inclusion criteria were consistent with the study objectives, aiming at health professionals in specific units, while the exclusion criteria were clearly defined to ensure the safety of the participants. However, the lack of a control group limited the capacity to fully evaluate its efficacy, although a pre and post intervention comparison was performed to analyze the results.

This study reaffirms the importance of multidisciplinary action, involving professionals from different areas to promote health more effectively in the workplace. The integrated approach allows for a comprehensive attention to needs, and it is essential that long-term studies are carried out to assess the sustainability of the benefits observed and the maintenance of behavioral changes over time.

The study conducted by Geany *et al*, although directed only at the industrial sector, showed an effective form of intervention, through the Food Choice at Work program. Intervening directly in the type of food supplied at the companies and in the education of the employees presented significant results as to the intake of salt and fat, improving indices such as body mass.

Matsugaki *et al* compared the intervention and control groups, statistically significant differences were found in various outcomes such as aerobic aptitude, muscle strength, cholesterol levels and symptoms of depression, revealing the positivity of the supervised interventions. Some studies have already shown the positive impact of physical exercise on working conditions and mental health [31].

In another study with nurses [27] directed at food and physical activity, despite the low adherence, the results indicated positive variables such as a significant increase in the consumption of fruits and vegetables. On the other hand, a reduction in physical activity was noted, raising doubts about the general efficacy of the intervention to promote changes of behavior in this population.

In the SHIELD program, the authors showed positive effects over the short and long term, a perception that change was sustained over time, with a notable increase of fruit and vegetable consumption, and the reduction of smoking and alcohol use after 24 months. This study provides a significant contribution which is its long term analysis, demonstrating that it is possible that the positive changes observed in the beginning will be maintained over time [28].

Most of the evidence found suggests that interventions to promote health in the workplace are a global concern [32]. Dietary interventions that consider adjusting meals to improve the synchronization of the biological clock may be more effective. In addition, regular physical activity, especially at strategic times, has the potential to mitigate the consequences of shift work, helping to restore glycemic balance and improve energy metabolism. Understanding these mechanisms can provide a more solid basis for developing collective intervention strategies with the potential to improve health in the long term. Interventions such as improving the quality of food available in work environments, including nutrient-rich options with a low glycemic index, can help reduce the negative effects of circadian dysregulation on metabolism.

It is essential that specific organizational policies are implemented to guarantee the effectiveness of the strategies. The literature suggests that changes in organizational practices, the organization of working hours and schedules, the provision of healthier food and the incorporation of regular breaks for physical activity, can significantly improve workers' well-being. In addition, implementing more flexible shift schedules can reduce the risk of chronic exhaustion and work-related illnesses, such as cardiovascular problems and sleep disorders. Public policies implemented with a focus on exercise breaks, a balanced diet and mental health support programs have also shown positive results, such as reduced absenteeism and increased productivity. These recommendations are crucial for turning scientific theory into practices that can be effectively applied in the workplace.

Studies show that strategies such as the use of accessible technologies, multidisciplinary health programs, dietary interventions and encouraging regular physical activity are key to improving workers' well-being and promoting health. The use of technology is an innovative strategy that stands out for its accessibility, practicality and feasibility, especially in populations with irregular working hours. By integrating a targeted text messaging system, the intervention adapts to the routine of these workers, overcoming common barriers in traditional health programs, such as lack of time or difficulty in accessing face-to-face health services. Technology acts as an effective communication channel, but also plays a crucial role in motivating and providing ongoing support for adherence to the behavioral change program. Despite the favorable results found, the limitations pointed out, such as the loss of follow-up and the difficulty of maintaining long-term adherence, show that occupational health interventions require continuous investment in human and material resources. The multi-professional team cannot act in isolation; investment in equipment and educational resources is needed to ensure effectiveness.

The review covers different types of study design, including randomized clinical trials (RCTs), retrospective studies and pilot interventions. The diversity in study designs offers a broad perspective, but the variability in research methods also brings significant challenges for comparability, which justifies the decision not to carry out a meta-analysis. In addition, most studies focus on short-term interventions and outcomes, which may not adequately reflect the long-term health benefits or risks associated with shift work. Considering that the metabolic and health effects of shift work can accumulate over time, assessing the long-term sustainability of these interventions is essential and is limited in the available literature.

## Conclusions

The diversity of interventions suggests that there is not only a single solution to promote health at the workplace. The strategies may be adapted to the specific needs and contexts of the workers and work environments. The adherence of managers is a way to reinforce the importance of preventive actions and allow a better adaptation of the organizational context to these activities. In all studies, the combination of the quantitative and qualitative approaches in the evaluation

of the process provides a greater understanding of the perceptions of the participants and of the lessons learned for future research and interventions. However, it is essential to approach the limitations, such as loss of follow up and unexpected challenges to improve the applicability and dissemination of the results. Future research should focus on comparing day and night workers and how the involvement of managers contributes to the profile and maintenance of results.

## Supporting information

**S1 Data. Dataset containing all studies identified in the systematic review search, including those included in the analysis and those excluded, with information on selection criteria.** This work was supported by Research Financing and Incen-tive (Fipe) from Hospital de Clínicas de Porto Alegre (HCPA).
(DOCX)

## Author contributions

**Conceptualization:** Francielle Lopes dos Reis, Ticiana da Costa Rodrigues, Maria Carlota Borba Brum.

**Data curation:** Francielle Lopes dos Reis, Julio Cesar Ferreira Bertoloto, Sheila de Castro Cardoso Toniasso, Camila Pereira Baldin, Maria Carlota Borba Brum.

**Formal analysis:** Francielle Lopes dos Reis, Julio Cesar Ferreira Bertoloto, Sheila de Castro Cardoso Toniasso, Maria Carlota Borba Brum.

**Funding acquisition:** Maria Carlota Borba Brum.

**Investigation:** Francielle Lopes dos Reis.

**Methodology:** Francielle Lopes dos Reis, Maria Carlota Borba Brum.

**Supervision:** Ticiana da Costa Rodrigues.

**Validation:** Francielle Lopes dos Reis, Maria Carlota Borba Brum.

**Visualization:** Francielle Lopes dos Reis, Camila Pereira Baldin.

**Writing – original draft:** Francielle Lopes dos Reis, Julio Cesar Ferreira Bertoloto, Juliana Barros Rodrigues, Dvora Joveleviths, Maria Carlota Borba Brum.

**Writing – review & editing:** Ticiana da Costa Rodrigues, Sheila de Castro Cardoso Toniasso, Camila Pereira Baldin, Juliana Barros Rodrigues, Dvora Joveleviths, Maria Carlota Borba Brum.

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
