## [Decision Letter · Decision Letter 0]

PONE-D-24-31823The impact of interventions in the workplace promoting exercise and a healthy diet among shift workers: a systematic reviewPLOS ONE

Dear Dr. dos Reis,

Thank you for submitting your manuscript to PLOS ONE. After careful consideration, we feel that it has merit but does not fully meet PLOS ONE’s publication criteria as it currently stands. Therefore, we invite you to submit a revised version of the manuscript that addresses the points raised during the review process.

We look forward to receiving your revised manuscript.

Kind regards,

António Raposo

Academic Editor

PLOS ONE

Journal Requirements:

“Research and Events Incentive Fund - FIPE from Hospital de Clínicas de Porto Alegre (2022-0618)’

5. We note you have included a table to which you do not refer in the text of your manuscript. Please ensure that you refer to Table 3 in your text; if accepted, production will need this reference to link the reader to the Table.

6. As required by our policy on Data Availability, please ensure your manuscript or supplementary information includes the following:

Additional Editor Comments:

The authors should consider revising their manuscript according to all the reviewers' comments.

Reviewers' comments:

Reviewer's Responses to Questions

**Comments to the Author**

1. Is the manuscript technically sound, and do the data support the conclusions?

Reviewer #1: Partly

Reviewer #2: Yes

Reviewer #3: Yes

Reviewer #4: Yes

2. Has the statistical analysis been performed appropriately and rigorously? 

Reviewer #1: N/A

Reviewer #2: Yes

Reviewer #3: I Don't Know

Reviewer #4: Yes

3. Have the authors made all data underlying the findings in their manuscript fully available?

Reviewer #1: Yes

Reviewer #2: Yes

Reviewer #3: Yes

Reviewer #4: Yes

4. Is the manuscript presented in an intelligible fashion and written in standard English?

Reviewer #1: Yes

Reviewer #2: Yes

Reviewer #3: Yes

Reviewer #4: Yes

5. Review Comments to the Author

Reviewer #1: Comments:

The manuscript addresses a valid research question within the journal's scope. Additionally, it explores an important issue regarding the impact of interventions promoting healthy eating and/or physical activity among shift workers.

The title accurately reflects the manuscript's focus and is concise, while the abstract precisely describes the objectives, methods, results, and conclusions.

To improve the structure and flow of the manuscript, I would like to offer some comments, questions, and suggestions.

Abstract:

- Provide protocol registration information;

- Indicate how risk of bias was assessed.

Introduction

- Define the WHO abbreviation upon first mention.

Materials and Methods

- Define the PICO strategy, clarifying the Population, Intervention, Control, and Outcomes beyond just identifying the central question;

- It is unnecessary to provide dates in parentheses; I suggest including the dates in the just main text instead;

- Additionally, phrases like "(for instance, sex)" or "(for instance, number of employees)" and similar citations are unclear. It is ambiguous whether only the items in parentheses are considered in the studies or whether additional data are included. I suggest rephrasing: "No limitations were established regarding the gender of the subject, workplace size, number of employees, and [list additional items]."

Exclusion Criteria

- Were other study types, such as general reviews (narrative, systematic with or without meta-analysis), preclinical studies, ecological studies, and cost-effectiveness analyses, included? If not, consider stating this explicitly;

- In this section, it states that "studies on youths less than 18 years old" were excluded, yet in the previous section, it mentions "No limitations were established regarding the characteristics of the subject (for instance, sex)." I suggest specifying "evaluated individuals aged ≥ 18 years" to clarify this in the eligibility criteria.

Evaluation of Bias

- What about studies that were not RCTs? How were they assessed? I suggest adding an additional evaluation method for them.

Results

Table 1

- Standardize table borders to the same thickness; for clarity, typically only horizontal lines are used.

Line 1: Describe the exclusion criterion as with other lines; if not already mentioned, please specify;

- Line 2: Describe the inclusion criterion similarly; if not already mentioned, please specify;

- Consider using abbreviations only within the table and including descriptions in a line following the table as a caption;

- Move the Author/Year column to the first position, and instead of placing the reference in superscript with the title, insert the reference in square brackets next to the author's name and publication year, as specified in the author’s guide. For example, in the first line: Monnaat Sie, M et al., 2023 [11];

- Only the last name of the first author is necessary, and a semicolon should follow "et al.," Please review and reformat accordingly.

Table 2

- Format the borders similarly to Table 1;

- Include references in square brackets in all rows in column 1;

- Format and standardize italicized items in line 1;

- In the study by Torquati L et al., 2017, correct the text in column 4: "post-intervention anda t 6 months.";

- What are the asterisks in the results in line 1? Please explain;

- Use abbreviations in the table text, with descriptions in a caption for terms such as BMI and FFQ;

- Adjust cell indentation; some spaces are misaligned;

- Specify whether "duration" refers to the intervention or study period to avoid confusion for readers.

First Paragraph of Results Presentation

- Consider adding basic data on sample size (n) and percentages, average age, gender proportions, commonly represented countries, and average duration;

- Clarify "RE-AIM structure."

Line 29, Regarding the Assessment Tool, RoB 2.0

- For studies where this method was not applied, an alternative risk-of-bias assessment tool should be included so all studies undergo bias evaluation;

- Provide a general conclusion of this analysis, classifying the overall risk of bias in your study as low, medium, or high; this information would be beneficial.

Discussion

- The discussion currently presents each study's findings individually; this section should contextualize and connect these findings with the existing literature, incorporating recent references. I recommend restructuring this section;

- Consider presenting these results more succinctly within the results section and using this section to explore common themes that warrant further discussion.

- Include a section on study limitations.

Conclusion

- Highlight key findings and potential areas for further research.

Line Numbering

- Line numbering should be sequential and continuous, without restarting on each page. I suggest correcting this.

References

- I recommend standardizing references throughout the text as specified in the author's guide, using square bracket numbers.

Reviewer #2: The article addresses a highly important and current topic concerning the effects of interventions that promote healthy eating habits and physical activity in the workplace, particularly among shift workers. The systematic analysis of the available literature offers a valuable source of knowledge, with the authors effectively presenting the findings of individual studies, clearly illustrating the positive effects of the interventions undertaken.

However, the article’s introduction requires certain additions. It is currently somewhat brief, and while it outlines the main points, it lacks several essential elements that could better justify the importance of the analysis undertaken. The authors could more extensively describe the physiological mechanisms underlying the effects of shift work on metabolic health. A brief discussion of how disruptions to circadian rhythms affect glucose metabolism, increase the risk of insulin resistance, and alter the secretion of hormones responsible for hunger regulation (e.g., leptin and ghrelin) would better illustrate the scale of health challenges associated with shift work. In the literature review on the impact of shift work on health, it would also be useful to include an overview of existing studies and their limitations, which would better justify the need for a systematic review. It would be worthwhile to indicate what gaps exist in the current literature and what questions remain unanswered. Furthermore, the introduction should expand on the role of the workplace as an environment conducive to health promotion, highlighting why the workplace - due to time spent there and opportunities to implement health programs - is ideal for such interventions. The introduction would also benefit from a discussion of the potential health and economic benefits that effective interventions could bring, in terms of reducing healthcare costs, improving productivity, and enhancing employees' quality of life. This would add a practical dimension to the introduction, useful for both public health specialists and organizational managers.

Aside from these suggestions, the article presents a high level of academic merit. The authors have provided a detailed description of the systematic review methodology, making the process of study selection and the analysis conducted transparent and reliable. A diverse set of databases and well-defined inclusion and exclusion criteria were included, which further enhances the value of the analysis. The results are presented clearly and comprehensibly, allowing readers to easily follow the effects of the interventions analyzed and their impact on the health of shift workers.

Reviewer #3: This systematic review protocol outlines an effort to examine the impact of interventions promoting healthy eating and/or physical activity among shift workers.

The study mentions that no comparison was made between day and night shifts, which is a critical shortcoming. Night shift workers are especially vulnerable to metabolic issues because of disrupted circadian rhythms and glucose metabolism.

The review includes various study designs, such as randomized controlled trials (RCTs), retrospective studies, and pilot interventions. While diversity in study design can provide a broad perspective, the variability in study methods also introduces significant challenges for comparability.

Most studies likely focus on short-term interventions and outcomes, which may not reflect long-term health benefits or risks associated with shift work. Given that the metabolic and health effects of shift work accumulate over time, assessing the long-term sustainability of these interventions is crucial.

The review acknowledges that a variety of interventions were assessed, ranging from dietary changes to physical activity initiatives. While this breadth provides a comprehensive view, it also complicates direct comparisons between studies. Different interventions may have varying impacts on health outcomes, and generalizing the results across all interventions may overlook more effective or less effective strategies. Ideally, the review should stratify findings by intervention type (e.g., diet vs. exercise) to identify which approaches are most beneficial for specific groups of shift workers.

The manuscript highlights the role of managerial support in improving adherence but lacks a detailed exploration of the barriers and facilitators to worker adherence. Shift workers often face practical challenges, such as irregular work hours and sleep deprivation, which can hinder their ability to maintain healthy eating or exercise habits.

Manuscript could benefit from a deeper discussion of the underlying mechanisms by which dietary and physical activity interventions influence health outcomes in shift workers. Understanding the biological pathways involved, such as circadian disruptions affecting glucose metabolism, could help design more targeted interventions. Incorporating this mechanistic understanding would make the findings more scientifically robust and relevant.

The paper mentions the importance of managerial support but does not provide specific policy recommendations for organizations to adopt. Providing actionable insights for workplace policies, such as offering healthier food options during night shifts or scheduling exercise breaks, could enhance the practical utility of the study. Additionally, exploring organizational changes such as rotating shift schedules or offering flexible work hours might further promote health among shift workers. This should be included.

Reviewer #4: Need Major Revision. Here i add my comment please take a look. You are doing well in your article. Best wishes for your next writing and research.

Manuscript Summary: This study investigates workplace promoting exercise and a healthy diet among shift workers, using Systematic review protocol. The authors conclude that the managers are a way of reinforcing the importance of preventative actions and allows a better adaptation of the organizational context to these activities, there is no single solution. Overall, this study adds valuable insights into workers and their development strategies through exercise and healthy diets.

General Comments:

Contribution to the Field: The study addresses some overcoming context like exercise, and diet habits and its necessity. However, not provide any specific outcome like if this will be taken then it provides much more effective way.

Is the paper well written? Yes, very well written but some sections need much more clarity in accent use.

Specific Comments:

Introduction: In the introduction, the structure needs to be altered. Also, the research gap could be explained more.

Why particularly this day and night shift workers were selected could be explained? In the end of the introduction, how this study could help to solve the problem could be added.

What is the main question addressed by the research?

Methods: Is it relevant and interesting? Very interesting manuscript

How original is the topic? Many groups are studying this topic.

Figures/Tables: Is the text clear and easy to read? Yes, very interesting to read.

Conclusion: Well define.

Recommendations for Improvement:

Please include any additional comments on the tables and figures and quality of the data.The tables and figures are nicely formatted. The way they have added limitations and practical applications make this manuscript pretty unique and clear.

Additional comments:

1. What is the main question addressed by the research? The group is trying to applied for other group of worker like all types of worker? Write it more specifically.

2. What parts do you consider original or relevant for the field? What specific gap in the field does the paper address? A lot of research is going on this exercise and nutrition and its relevant.

3. What specific improvements should the authors consider regarding the methodology? What further study should be considered? Maybe a broader type of sample size of article may be including Scopus index articles.

Decision Recommendation: Major Revision.

6. PLOS authors have the option to publish the peer review history of their article (what does this mean? ). If published, this will include your full peer review and any attached files.

**Do you want your identity to be public for this peer review?** For information about this choice, including consent withdrawal, please see our Privacy Policy .

Reviewer #1: **Yes: ** Marcela Gomes Reis

Reviewer #2: No

Reviewer #3: **Yes: ** M. João Reis Lima

Reviewer #4: **Yes: ** Al Azim

---

## [Author Response · Author response to Decision Letter 1]

16 Apr 2025

First of all, we would like to thank you for the effort and time invested in this manuscript. We are certain that these are very valuable comments that highlight its quality. We had the opportunity to make improvements and adjustments to the manuscript and we are submitting them for evaluation. We would like to emphasize that no artificial intelligence was used in the preparation and review of the document. We will continue to respond point by point to the comments of the editor and reviewers.

Response to Editor Comments:

Comment 1

Response: Thank you for your comments. We agreed and have made the changes in the manuscript.

Comment 2:

Response: This information was reviewed when the manuscript was sent. Thank you.

Comment 3:

Response: The funder has participated by translating the manuscript and decision to publish.

Comment 4

Response: After a review of the selected articles, we noted that due to the different study designs, it was not possible to carry out the meta-analysis, we sent a spreadsheet with the studies found and those included in the search for the systematic review. Therefore, the requested data is not made available.

Comment 5: We note you have included a table to which you do not refer in the text of your manuscript. Please ensure that you refer to Table 3 in your text; if accepted, production will need this reference to link the reader to the Table.

Response: Thank you for your comment. This item has been reviewed and corrected in the manuscript.

Comment 6:

Response: This information was summarized and appended as supplementary information to the main text. All included studies are published. No adjustments were made for missing data because it was not possible to compare studies and, consequently, perform a meta-analysis. Supporting information was obtained only from selected studies.

The data from the studies used are included in the attached supplementary information file.

Risk of bias was assessed, and information is provided in tables 3 and 4 in the text of the article.

Response to Reviewers

Reviewer #1:

Response: Thank you for your comments. We agreed and have made the changes in lines 35 to 39.

Comment2:

Response: Thank you for your comments. We agreed and have made the changes in line 69.

Comment 3

Response: Thank you for your comments. We agreed and have made the changes in lines 130 to 135 and 142 to 144.

Exclusion Criteria

Comment 4:

Response: Thank you for your comments. The description of the inclusion and exclusion criteria has been changed in lines 151-152.

Comment 5:

Response: Thank you for your comments. Added suggested method for the studies in question in lines 207-208.

Results

Comment 6:

Response: Thank you for your comments. We agreed and have made the changes in table 1.

Comment 7: Response: Thank you for your comments. We agreed and have made the changes in table 2.

Comment 8: Response: Thank you for your comments. We agreed and have made the changes in lines 222 to 239, and 259-260.

Comment 9:

Response: Thank you for your comments. We agreed and have made the changes..

Comment 10:

Response: Thank you for your comments. We chose to assess the risk of bias in the articles using the ROBIN I tool.

Comment 11:

Response: Thank you for your comments. The data were reviewed and we added the ROBIN I assessment method as suggested.

Comment 12:

Response: Thank you for your comments, they were essential for restructuring the discussion. We agreed and have made the changes in lines 316 to 356 and 361 to 365 and 371 to 418.

Comment 13:

Response: Thank you for your comments. We included this paragraph in the conclusion “Future research should focus on comparing day and night workers and how the involvement of managers contributes to the profile and maintenance of results”.

Comment 14

Response: We agreed and have made the changes in the line numbering and the references.

Reviewer #2:

Comment 1:

Response: Thank you for your comments. We included this paragraph in the introduction “Blood glucose homeostasis is associated with central circadian rhythmicity as well as peripheral oscillators. During periods of activity, blood sugar levels are determined mainly by nutrient intake. During rest and fasting, endogenous hepatic glucose production occurs and glucose levels are maintained within a relatively narrow range. [8]. This also includes intestinal hormones, associated with the regulation of hunger and satiety, as well as gastrointestinal functioning and the kinetic properties of the stomach. Ghrelin stimulates food intake, while leptin promotes satiety. The production of these hormones can be interfered with by sleep deprivation and food intake, leading to an internal desynchronization of the circadian rhythmicity of satiety-related peptides, especially leptin, causing an imbalance between energy intake and expenditure [9-10-11-12]”.

Comment 2:

Response: Thank you for your comments. We included this paragraph in the introduction “Numerous opportunities exist for health promotion and chronic disease prevention initiatives in the workplace, including (a) health screening, (b) tobacco cessation activities, (c) the promotion of healthy food choices and weight loss, (d) physical activity, and (e) routine vaccinations [16]. However, the efficacy of these interventions has not yet been fully established, and it is necessary to perform a broad review of the evidence available to evaluate whether the interventions present favorable results regarding the living habits adopted by the workers, especially when it is a matter of diet and exercises essential to prevent NCDs”.

Comment 3:

Response: Thank you for your comments. We included this paragraph in the introduction “Workplace interventions have objectives related to the promotion and reduction of health care costs. In this sense, studies have been carried out to identify the effectiveness of these interventions, showing positively significant effects in relation to physical activity and nutrition, related to self-reported behavior, but without significant results in relation to physical health measures, rates of medical diagnoses or use of health services [17]”.

Comment 4:

Response: Thank you for your comments. We included this paragraph in the introduction “There is evidence of the benefits of supporting health promotion activities in the workplace with the aim of maintaining the capacity, health and quality of life of workers, with potential impacts on health care costs and on the productivity, competitiveness and sustainability of companies, communities and even national economies, considering both the risks in work environments and other factors such as ageing and the need to adapt the workplace of people with chronic conditions [15]”.

Comment 5

Response: Thank you for your comments.

Reviewer #3:

Comment 1:

Response: The aim of the study is to examine the impact of interventions that promote healthy eating and/or physical activity among shift workers because night workers are more vulnerable. Unfortunately, in a search prior to the design of the systematic review protocol, we found no articles comparing workers from both shifts.

Comment 2:

Response: We agree with the reviewer's comment. Our search did not make it possible to compare the studies due to their different designs.

Comment 3:

Response: We agree with the reviewer's comment. We have included information about the follow-up of participants in the studies that conducted these evaluations in the discussion section of the manuscript. We believe that future studies should include instruments to assess adherence to behavioral changes and the medium- and long-term effects of health promotion interventions among night workers. In this sense, the involvement of managers can have a positive impact on both perspectives.

Comment 4:

Response: In reality, there is no single strategy considered beneficial when it comes to diet and physical activity. A combination of the two, in fact, would be the ideal approach. However, in studies it is difficult to make a direct comparison as they generally focus on one of the strategies alone or on the combination of both showing favorable results.

Comment 5:

Response: Thank you for your comments. Only in the study conducted by Monnaatsie, the facilitating factors or barriers to the implementation and maintenance of the program to promote physical health were evaluated. We agreed and have made the changes in discussion.

Comment 6:

Response: Thank you for your comments. We agreed and have made the changes in discussion.

Comment 7:

Response: Thank you for your comments. We agreed and have made the changes in discussion in lines 371 to 418.

Reviewer #4:

General Comments:

Comment 1:

Response: Thank you for your comments.

Comment 2:

Response: Thank you for your comments. The text has been reviewed and changes made.

Specific Comments:

Comment 3:

Response: Thank you for your comments. We agreed and have made the changes in the introduction, as highlighted in the text.

Comment 4:

Response: Our aim was to summarize and evaluate the evidence, with a view to gaining insights on which to base the development of effective policies and practices to improve the health and well-being of professionals.

Comment 5:

Response: To analyze existing studies on workplace interventions aimed at promoting physical exercise and healthy eating among shift workers and evaluate their effectiveness

Recommendations for Improvement:

Comment 6:

Response: Thank you for your comments. The tables and figures have been reviewed and changes made.

Additional comments:

Comment 1

Response: To analyze existing studies on workplace interventions aimed at promoting physical exercise and healthy eating among shift workers and evaluate their efficacy. Our research group is developing a project with health promotion interventions focused on healthy eating, physical activity, sleep quality and well-being among health professionals at a university hospital in southern Brazil in 2023.

Comment 2.

Response: Research involving evaluation and monitoring of health interventions aimed at night or shift workers is scarce. In addition to the complexity of their implementation, there is a diversity of occupational activities involved and wide variation in the organization of work shifts (duration, work breaks, rotation and rest periods), making it difficult to define a single protocol for these workers. This review highlights the importance of these actions being part of a specific organizational policy, using different strategies such as the use of accessible technologies and multidisciplinary health programs, and the planning of strategies and individual objectives associated with nutritional and physical activity support, addressed individually or in groups, demonstrate that offering information on the topics and resources for changes in participants' lifestyles has positive results, which extend beyond the intervention period.

Comment 3.

Response: We consider that the methodology used was appropriate for the objectives and we agree with the reviewer that the use of the Scopus index tool can qualify the search for articles relevant to the research question, however, we have no experience with the use of the tool and chose not to include it in the study methodology.

---

## [Decision Letter · Decision Letter 1]

The efficacy of interventions in the workplace promoting exercise and a healthy diet among shift workers: a systematic review

PONE-D-24-31823R1

Dear Dr. dos Reis,

We’re pleased to inform you that your manuscript has been judged scientifically suitable for publication and will be formally accepted for publication once it meets all outstanding technical requirements.

Kind regards,

António Raposo

Academic Editor

PLOS ONE

Additional Editor Comments (optional):

Reviewers' comments:

Reviewer's Responses to Questions

**Comments to the Author**

1. If the authors have adequately addressed your comments raised in a previous round of review and you feel that this manuscript is now acceptable for publication, you may indicate that here to bypass the “Comments to the Author” section, enter your conflict of interest statement in the “Confidential to Editor” section, and submit your "Accept" recommendation.

Reviewer #2: All comments have been addressed

Reviewer #3: All comments have been addressed

Reviewer #4: All comments have been addressed

2. Is the manuscript technically sound, and do the data support the conclusions?

Reviewer #2: Yes

Reviewer #3: Yes

Reviewer #4: Yes

3. Has the statistical analysis been performed appropriately and rigorously? 

Reviewer #2: N/A

Reviewer #3: Yes

Reviewer #4: Yes

4. Have the authors made all data underlying the findings in their manuscript fully available?

Reviewer #2: Yes

Reviewer #3: Yes

Reviewer #4: (No Response)

5. Is the manuscript presented in an intelligible fashion and written in standard English?

Reviewer #2: Yes

Reviewer #3: Yes

Reviewer #4: Yes

6. Review Comments to the Author

Reviewer #2: Thank you for making the revisions to your manuscript. The changes you made have improved the paper, and I appreciate your efforts.

Reviewer #3: Considering the major improvements made in the paper, I think that the manuscript can now be published.

Reviewer #4: (No Response)

7. PLOS authors have the option to publish the peer review history of their article (what does this mean? ). If published, this will include your full peer review and any attached files.

**Do you want your identity to be public for this peer review?** For information about this choice, including consent withdrawal, please see our Privacy Policy .

Reviewer #2: No

Reviewer #3: **Yes: ** M. João Reis Lima

Reviewer #4: **Yes: ** Al Azim

---

## [Editor Report · Acceptance letter]

PONE-D-24-31823R1

PLOS ONE

Dear Dr. dos Reis,

I'm pleased to inform you that your manuscript has been deemed suitable for publication in PLOS ONE. Congratulations! Your manuscript is now being handed over to our production team.

Kind regards,

on behalf of

Dr. António Raposo

Academic Editor

PLOS ONE